# Identification of Aluminothermic Reaction and Molten Aluminum Level through Vision System

**DOI:** 10.3390/s23125506

**Published:** 2023-06-12

**Authors:** Yuvan Sathya Ravi, Fabio Conti, Paolo Fasoli, Emanuele Della Bosca, Maurizio Colombo, Andrea Mazzoleni, Marco Tarabini

**Affiliations:** 1Department of Mechanical Engineering, Politecnico di Milano, 20156 Milan, Italy; yuvansathya.ravi@polimi.it (Y.S.R.);; 2One-Off Innovation, 23900 Lecco, Italy

**Keywords:** aluminothermic reaction, level measurement, flame recognition, line detection, aluminum production, vision system, image processing

## Abstract

During the secondary production of aluminum, upon melting the scrap in a furnace, there is the possibility of developing an aluminothermic reaction, which produces oxides in the molten metal bath. Aluminum oxides must be identified and removed from the bath, as they modify the chemical composition and reduce the purity of the product. Furthermore, accurate measurement of molten aluminum level in a casting furnace is crucial to obtain an optimal liquid metal flow rate which influences the final product quality and process efficiency. This paper proposes methods for the identification of aluminothermic reactions and molten aluminum levels in aluminum furnaces. An RGB Camera was used to acquire video from the furnace interior, and computer vision algorithms were developed to identify the aluminothermic reaction and melt level. The algorithms were developed to process the image frames of video acquired from the furnace. Results showed that the proposed system allowed the online identification of the aluminothermic reaction and the molten aluminum level present inside the furnace at a computation time of 0.7 s and 0.4 s per frame, respectively. The advantages and limitations of the different algorithms are presented and discussed.

## 1. Introduction

Aluminum is the second most available metallic element on earth and is one of the most used metallic elements in industrial applications [1,2]. It is one of the most recyclable materials, and nearly 75% of all the aluminum produced is still in use today. The process of aluminum recycling requires only 5% of the energy consumed to produce primary aluminum, its extraction being extremely energy-consuming [1]. Thanks to its resistance to corrosion, non-toxicity, and high thermal and electrical conductivity, aluminum is largely used in the majority of industrial sectors [2].

When melted in furnaces during secondary production, a fraction of the aluminum reacts with oxygen from the surrounding environment and forms oxides, commonly referred to as “*dross*” [3,4,5]. Oxides can form because of several reasons; the one that is of interest for this study is the dross obtained after the aluminum thermite reaction, also known as the *aluminothermic reaction*, an exothermic process that leads to temperatures often exceeding 1500 °C [3]. The aluminothermic reaction is characterized by peculiar fire flames, which are extremely bright and burst in spots on the melted surface. This phenomenon implies a rapid chemical reaction of aluminum with oxygen, nitrogen, and carbon dioxide to form different compounds, including dross [3,6,7].

Considering that the value of aluminum is much higher than the one of oxides, it is important in terms of operating costs and energy consumption to minimize the amount of dross. Currently, in industrial practice, the presence of aluminothermic flames inside a melting furnace is identified manually through a visual inspection carried out by a skilled operator. The dross formed on the molten metal layer is removed by a “*skimming operation*”, where the molten layer is skimmed preventively or while the reaction is in progress [3]. Skimming is a manual process that demands the opening of the furnace, entailing a loss of thermal energy to the surrounding environment.

With the images acquired from a purposely designed camera system, aluminothermic flames present inside the furnace could be automatically identified [8,9,10,11,12,13,14,15,16]. The process requires distinguishing the aluminothermic flames from the ordinary flames that can be generated during the melting process, with images of inadequate quality, given the extreme environment in which the camera system operates. Flame recognition is a widely discussed topic in the literature. Flames are not homogeneous, as they might have different colors and shapes. Different studies highlight how changing the illumination of the scene under observation may affect the recognition capabilities of the algorithm [11,12]. Smoke generated by fire, or the presence of dust, can reduce the quality of the image, increasing the difficulty of flame recognition. Objects with colors like the ones of flames may be present in the image, which may lead to false positive recognitions [10,14]. Reflections can be another cause of false positives, occurring mainly in closed spaces. Appropriate strategies must be chosen to discard the undesired elements and obtain an optimal identification of the flames from the image. The known literature evidences that there are several technique*s* to identify flames in different scenarios, and the most coherent choice is to consider the specific context under analysis and exploit the simplifications it can provide [8]. Considering the case of aluminothermic flames in aluminum furnaces, the problem consists of the detection of a very bright element in a very dark environment. In this framework, the color of fire usually ranges from yellow and red to white, depending on the temperature and chemical reaction [12]. Therefore, color is intuitively the most evident static feature resulting in the wide use of different strategies [8,9,10,12,13,14,15]. This feature could be exploited using different color scales of the image, such as RGB (red, green, and blue), HSV (Hue Saturation Value), and grayscale.

Another problem in the continuous casting of aluminum arises from the necessity of controlling the temperature of the bath and the flow of liquid aluminum during the casting, since this parameter influences the crystalline microstructure of the final product [17]. The accurate measurement of molten aluminum level in the furnace is of paramount importance to obtain the optimal liquid metal flow rate, which influences the final product quality. Apart from the product quality, accurate online monitoring of the aluminum level helps avoid unscheduled stops of the plant operations, thereby improving the process efficiency.

One of the most common techniques for measuring the metal level is the use of sounding bars in which a metal rod made of steel is periodically immersed in the molten metal bath [18]. With this method, the melt level is determined by the difference in the color of the rod due to heat transfer. The resulting accuracy and repeatability are generally limited, and the operator’s safety is at risk, since the process is manual and is carried out by opening the furnace door. The application of other common contact methods for liquid level measurements, such as fiber Bragg grating [18,19], is critical due to the harsh environment inside the furnace, where the temperatures can exceed 1000 °C.

Several non-contact level measuring methods have been proposed by different authors; the sensing principles are based on γ-ray transmission [20,21], eddy current [22], induction [21], capacitance [23], and computer vision methods [20,24]. The use of γ-rays is not suitable in a production plant as it emits high-energy radiation, which can be harmful to the human body [20]. Measurement using inductance and capacitance [21,23] requires the sensors to be placed at a close distance to the melt surface, which is not possible inside a furnace. Similarly, the accuracy of electromagnetic methods based on Eddy current sensors [22] highly depends on the distance of the sensor from the measured object due to the attenuation of the electromagnetic field with distance. Computer vision has been used to measure the steel level in a tundish [20,24]. The method combines a sounding bar, a laser for triangulation, and a camera, although it requires a mechanical system to periodically deliver the bar, which does not make it a non-contact method. Other methods use a camera system to measure the level of liquids against a measuring graduated scale [25,26,27,28]. The use of a vision system provides advantages such as non-contact measurements, the absence of moving parts, and the possibility of providing online monitoring of the process. The key papers reviewed for this work from the literature are summarized in Table 1.

This research work aims to integrate different technologies existing in the literature for the automatic recognition of aluminothermic reaction and aluminum levels starting from the images acquired by a camera installed in aluminum melting and casting furnaces. The proposed methods and a case study of the obtained results are presented in this paper.

## 2. Method

### 2.1. Aluminothermic Reaction Identification Algorithms

Aluminum scrap is melted in the furnace using gas burners typically installed at the top, and aluminothermic reactions take place inside the furnace as the scrap is melted. Figure 1 and Figure 2 show sample images acquired from the furnace interior in which aluminothermic reaction takes place at different rates. Figures also show the possible different illumination conditions of the furnaces.

Two algorithms were developed to find the aluminothermic flames in the molten metal bath: the first is based on processing the image in color scale, while the second one analyses the image at grayscale level.

#### 2.1.1. Color Scale Method

The first algorithm is developed by exploiting the HSV and RGB color scales of the acquired video. Since the aluminothermic reactions occur on the surface of the molten metal bath, the Region of Interest (ROI) is limited to the bath surface, while the remaining part should be cropped. Two binary images (Mask 1 and Mask 2) can be obtained by masking a set of HSV and RGB filters on the ROI. The area in which the aluminothermic flames can be present will be identified with a mask that is the union of Mask 1 and Mask 2, following the workflow shown in Figure 3.

Mask 1 is obtained from the HSV scale, which provides information about the color intensity; the RGB cropped image is converted to an HSV frame and then binarized, imposing filter conditions on each channel (H, S, and V). The HSV parameters must be chosen with the aim of maximizing the possibility of identifying the pixels corresponding to aluminothermic flames and minimizing the identification of non-aluminothermic flame pixels. The hue is used as a first step to remove the pixels that do not belong to aluminothermic flames. Although complex procedures might be available, good results can be generally obtained with simple thresholds in the HSV plane, with an iterative process that aims to achieve a recognition rate close to unity and minimize the false positive identifications as well.

All the pixels that satisfy at least one of the HSV filter conditions were considered as possible aluminothermic flames and set as white. The remaining pixels were set as black. This process leads to a binary image with aluminothermic flame candidate pixels.

Mask 2 is obtained by imposing thresholds on the different RGB channels. As shown in Figure 1 and Figure 2, the burner flames have a different color with respect to the aluminothermic flames, and the RGB thresholds should be chosen to exclude the pixels related to burner flames. Additionally, in this case, a simple threshold on the RGB channels is sufficient in the defined ROI.

Mask 1 and Mask 2 are then merged to obtain the final mask in which the white pixels represent the ones with the identified aluminothermic flames.

#### 2.1.2. Grayscale Method

To reduce the computation time, an alternate method to identify the aluminothermic reaction was developed using the grayscale of the image. Since the aluminothermic flames are the brighter elements in the frame, a simple threshold typically allows us to identify the flames in the image.

### 2.2. Molten Aluminum Level Identification Algorithm

Inside the casting furnace, the molten aluminum surface and internal refractory walls have different light intensity values. The idea is to identify the line which is formed at the interface between the refractory wall and the melt surface. Utilizing the known knowledge about furnace dimensions and the relative position of the camera’s field of view, the melt level was quantified. Figure 4 depicts the workflow of the algorithm for each frame of the video.

#### 2.2.1. Image Pre-Processing

Figure 5 shows the images acquired from the furnace for level identification in two different operating conditions. In Figure 5a, the molten bath surface is at a lower level when compared with that in Figure 5b. The green rectangles highlight the area used for the identification of different operating conditions in terms of the melt level. Hence, the acquired image is cropped to analyze only the highlighted Region of Interest (ROI). The image dimensions were reduced from 2160 × 3840 to 600 × 750, thereby improving the processing time of the algorithm.

To further reduce computational effort and complexity, images are converted to a grayscale image and normalized. The histogram is then modified to improve the contrast; the first transformation consists of subtracting the minimum value of the range to each pixel of the starting image; the second operation consists of the multiplication of each pixel intensity for a gain computed as per Equation (1).
gain = 255/(max − min)(1)
where min and max are the minimum and maximum intensity values of the initial histogram, respectively. An example of the modified histogram is shown in Figure 6.

As edge detection techniques are affected by noise, reducing the noise components in the image is often useful; different noise-filtering techniques, such as median blurring, Gaussian blurring, and bilateral filtering, can be used; in our tests, the bilateral filter usually provided better results, as it always preserved the edge also upon varying the illumination conditions and melt levels.

#### 2.2.2. Reference System

The level is identified starting from the detection of a 3D reference system, as in ref. [29]. The furnace dimensions must be extracted from the furnace technical drawing. In the real world, the line separating the lateral wall and the ceiling is parallel to the line indicating the molten bath level and to the one separating the floor from the lateral wall. Camera perspective distortion makes these three lines converge to a vanishing point (Xv, Yv), whose coordinates can be determined during the camera calibration process with the empty furnace. Figure 7 illustrates the calibration of the reference system performed initially to compute the level of the molten metal bath.

#### 2.2.3. Level Detection

The level is detected using an edge detection process; we implemented two approaches based on the Canny edge detection and the Otsu method. Canny edge detection works based on gradient computation by looking for the local maxima of the gradient, whereas the Otsu method is an adaptive method that uses a threshold that minimizes the intra-class variance of the image histogram and divides the pixels into two classes. The idea is that the intensities of the pixels which belong to the edge are supposed to be very close to the threshold computed by the Otsu method. Methods were tested in different configurations, and the Canny method usually granted better results.

Hough Transform, a widely adopted line detection technique in the literature [30,31,32,33], is used to detect lines in binary images with detected edges. The method returns multiple lines, and the correct separating edge between the bath and the furnace is identified using geometrical considerations. The first criterion is that the identified line should be parallel with respect to the H line in the reference system (Figure 7). The second criterion is based on the melting dynamics of the aluminum inside the furnace. The molten metal level inside the furnace does not change rapidly. It changes slowly. Therefore, the algorithm keeps in memory the heights computed in *n* previous frames and their mean value. For the current frame, the line with computed height closer to the mean value of the previous *n* frames is chosen. With this set of criteria, a single line is selected, discarding the other lines. In Figure 7, the selected single line is marked in red.

## 3. Experiments

### 3.1. Identification of Camera Position

Two cameras were installed in two different furnaces: one for aluminothermic reaction identification in the melting furnace and the other for melt level identification in the casting furnace. The optimal position for the camera installation was chosen to maximize the molten bath surface in the image frame. We developed a MATLAB^®^ 2021-based raytracing algorithm to estimate the field of view (FoV) of the camera. The raytracing algorithm requires the following as inputs:Horizontal and vertical resolution of the camera sensor;Horizontal and vertical view angle of the camera sensor;Physical dimensions of the aluminum furnace;Position and orientation of the camera in the furnace.

The first 2 parameters are declared by the manufacturer, and the furnace dimensions depend on the specific application. Specific constraints can derive from the presence of structural elements in the furnace that prevents the installation of the camera in specific positions. Figure 8a,b represent the estimated field of view of the cameras for aluminothermic reaction identification and melt level identification, respectively. The blue dots represent the molten aluminum bath. The green box represents the area framed in the FoV of the camera. The red crosses on the molten bath are used to trace the real distances in the frame. Simulating the field of views for different possible positions, the optimal one to install the cameras was identified.

### 3.2. Camera Calibration

The main goals of the calibration procedure are the computation of the lens distortion coefficients and depth of field of the camera. In our case, a Canty ExtremeTemp camera (which can operate at high temperatures of up to 1260 °C) was selected (Figure 9). It features a CCD sensor with a resolution of 8 MP. The electronics of the camera system are protected with a NEMA 4/IP66 enclosure, which is separated from the process by a fused, glass-to-metal hermetic seal to separate the camera electronics from the harsh environment. The system is equipped with an efficient air-cooling system, which can lower the temperature of the camera electronics and the optics.

The setup to perform the calibration consists of a checkerboard calibration system and a fit-for-purpose illumination technique. A checkerboard (with size 600 mm × 840 mm, square side 60 mm) was used as a calibrator. The calibrator was placed at a distance that was equal to the *focal length* of the camera. The calibrator was illuminated by a lighting system with 6000-lumen lamp. The formula used to calculate the theoretical *focal length* is given in Equation (2):(2)$$Focal\;length\;=\;\frac{z\;installation\;-\;z\;molten\;bath\;level}{\left|\sin\alpha \;\times \;\cos\beta \right|}$$
where *α* is the angle related to the downward rotation, *β* is the angle related to the rightward rotation, and *z* is the height (in mm) with respect to the bottom of the furnace. During the calibration process, a set of images were focused and acquired. The acquired images were given as inputs to the MATLAB^®^ 2021 Camera Calibration Toolbox. The distortion coefficients of the lens and depth of field were then computed. The calibrator was moved closer and farther with respect to the computed *focal length* to see the depth in terms of spatial distance at which the image remains sharp in focus.

## 4. Case Study

The case study described in this paper considers some images acquired from the furnace as a reference and analyzes the performance of the developed aluminothermic reaction identification methods and melt level identification method in different scenarios. The results of the developed algorithms and their performances are discussed in this section. For this case study, videos were acquired from the camera streaming 10 frames per second. Image processing was performed on one sample every 10 frames, and the processed image frames were looped over to display the result identification video.

### 4.1. Aluminothermic Reaction Identification Algorithms

The performances of the color scale and grayscale algorithms were compared in different conditions. Aluminothermic flames have different shapes and sizes depending on the rate of the reaction. So, one frame each for low, medium, and high rates of aluminothermic reaction was chosen as a reference. Figure 10 shows the identification obtained using color scale and grayscale algorithms for low- and medium-rate aluminothermic reactions. Identified aluminothermic flames are marked with green circles.

When the reaction occurs at a low to medium rate, both the color scale and grayscale algorithm were able to identify the aluminothermic flames correctly. They were also able to neglect the burner flames from the identification. Figure 11 shows the identification obtained for a frame with a high reaction rate. In this case, the color scale method was able to identify the aluminothermic flames clearly (Figure 11a), whereas the grayscale method was not able to provide an identification (Figure 11b).

Figure 12 shows a condition in which the illumination is low. The aluminothermic reaction is at a starting stage, and hence the flame is small. Both the color scale and grayscale methods provided accurate identifications of the aluminothermic flame in a low-light environment as well.

Overall, the color scale method provides comparatively better identification of aluminothermic flames for most of the cases analyzed. The optimal combination of filters on the color channels depends on the specific case under analysis.

#### Performance Comparision

Performances of the color scale and grayscale identification algorithms were compared based on the metrics’ average recognition rate and average cycle time. The average recognition rate gives an indication of how many pixels related to aluminothermic flames were identified correctly with respect to a reference frame. In an ideal case, the recognition rate is 1. The higher the number of unidentified aluminothermic flame pixels, the higher the value of this metric is. The average cycle time represents the average time taken by the algorithm to process a single image frame. This metric could be used to compare the computation speed of the algorithms. The results of the performance metrics are reported in Table 2.

Comparing the performances of the two methods, the color scale algorithm performs better in terms of recognition rate since its value (1.99) indicates that its identification is closer to reference (1) for the analyzed frames. The grayscale algorithm has a similar recognition rate to the color scale algorithm when the aluminothermic reaction starts at a low rate and grows slowly. However, as the aluminothermic reaction evolves to a high rate, the grayscale algorithm was able to identify fewer aluminothermic flame pixels. Hence, the recognition rate of the grayscale algorithm (6.51) is far from the reference. However, the advantage of the grayscale algorithm is that it has a much lower average cycle time (0.18 s) when compared with that of the color scale algorithm (0.69 s). Hence, the grayscale algorithm provides a faster identification.

In a common scenario where the aluminothermic reaction may occur at a low to medium rate, the developed algorithms are able to identify the aluminothermic reaction flames correctly. Even though the color scale method is computationally heavier than the grayscale method, its execution time is low enough (0.69 s) to be used to process a streaming video for online identification in the production plant.

### 4.2. Molten Aluminum Level Identification Algorithm

The results of the line identification algorithm for a lower and a higher molten aluminum level are shown in Figure 13a,b, respectively. The algorithm was able to identify the line corresponding to the molten aluminum level and compute its corresponding height inside the furnace.

The measurement repeatability was evaluated by observing the bath level and evaluating the measurement dispersion (Figure 14). A total of 450 consecutive image frames were taken for the observation. Generally, the melt level does not have a drastic change in a short time period. Hence, the measurements observed were around the same interval. The standard deviation of measurements was 2 mm with a mean of 756 mm.

#### Dynamic Performance

To test the dynamic performance of the level identification algorithm, a video was acquired from the furnace in which the molten aluminum level rises over time, and there is a change in illumination conditions inside the furnace for a certain time period as well. The window of time where the illumination conditions change inside the furnace is marked in purple (Figure 15). The time period marked in blue is at the end of the video, in which the burners were turned off and the furnace door was opened.

As observed in Figure 15, when there is a change in illumination conditions, even though the measurements are sparse, they are coherent with the trend of measurements in the other intervals. Therefore, it could be said that even though the algorithm is sensitive to changes in illumination conditions, it is robust enough to provide level measurements that are close enough to those of a good illumination condition. In the last part of the video, the burners were shut down, due to which the frame was completely dark. Hence, the level measurement algorithm struggled in that condition. Overall, the algorithm is robust enough to measure the level in a dynamic environment where the melt level and illumination conditions change over time. Computing the average execution times of 898 frames, the algorithm’s mean processing time of a frame was found to be 0.37 s, which is fast enough that it can be used for online identification.

## 5. Conclusions

Different computer vision algorithms were developed to identify the presence of aluminothermic reactions in the melting furnace and identify the molten aluminum level in the casting furnace. Comparing the color scale and grayscale methods for aluminothermic reaction identification, it was found that the color scale algorithm performs better as it is able to provide a recognition rate close to unity and was able to identify the aluminothermic flames in most of the frames analyzed, even when the reaction occurs at a high rate. With both methods, it was possible to exclude the identification of burner flames and identify only the flames caused by the aluminothermic reactions during the melting process. The algorithms were able to process each frame at a speed of less than 0.7 s, making it suitable for online processing of the video streamed from the melting furnace.

The proposed molten aluminum level measurement algorithm is able to provide reliable measurements in different operating conditions. When testing the level height measurement for 450 image frames, a standard deviation of 2 mm was observed. A faster processing time of 0.4 s per frame allows the online identification of the molten aluminum level present inside the furnace. It could be concluded that the proposed methods are able to provide an online identification of aluminothermic reactions and aluminum melt levels through a vision system installed in the aluminum furnace.

## Figures and Tables

**Figure 1 sensors-23-05506-f001:**
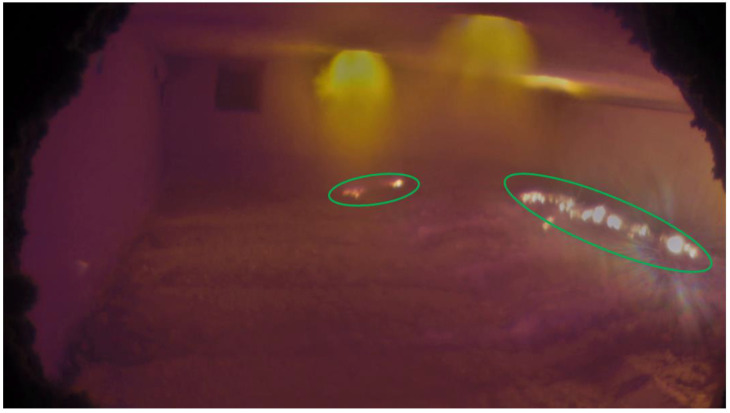
Image obtained from the inside of furnace. Aluminothermic flames are marked in green. Orange-colored flames on the roof are those from the burners.

**Figure 2 sensors-23-05506-f002:**
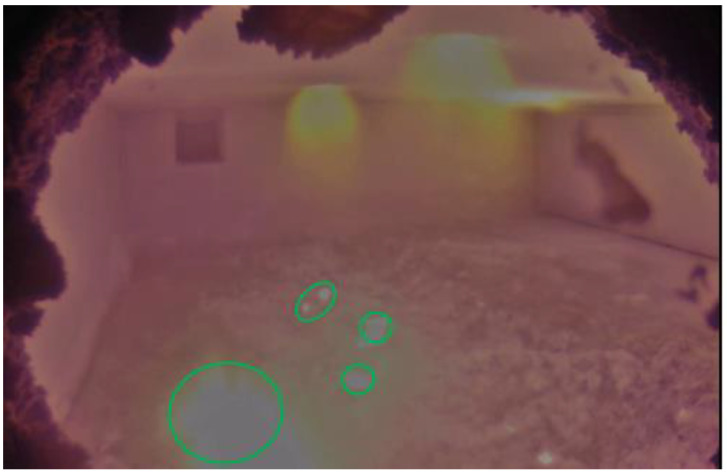
Picture showing widely developed aluminothermic flames on the molten bath. The aluminothermic flames are marked in green.

**Figure 3 sensors-23-05506-f003:**
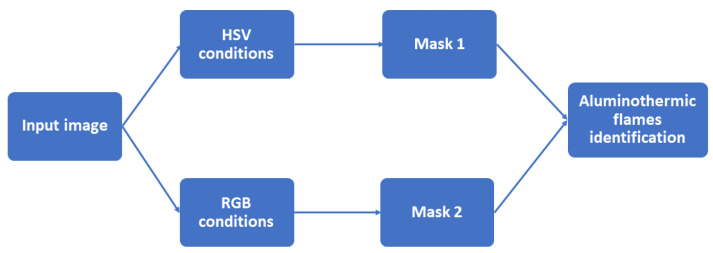
Workflow of color scale algorithm for each frame. White pixels in the resulting image represent the identified aluminothermic flames.

**Figure 4 sensors-23-05506-f004:**

Workflow of melt level identification algorithm.

**Figure 5 sensors-23-05506-f005:**
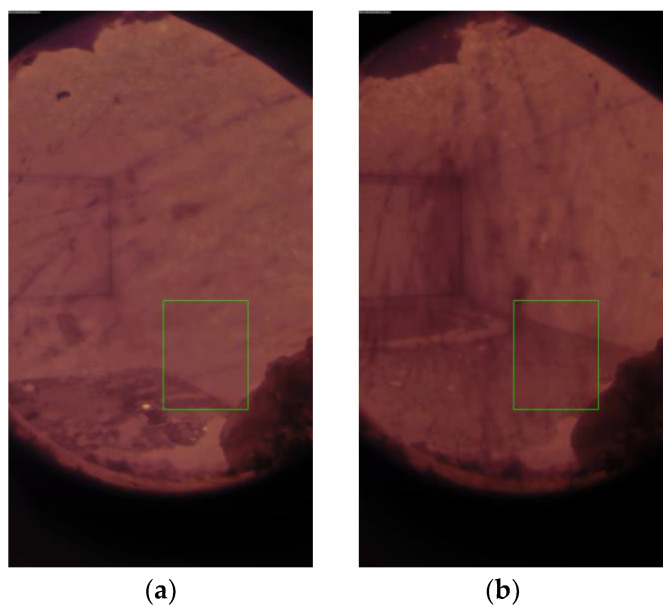
Images acquired from the casting furnace. Green box represents the Region of Interest. (**a**) Low molten aluminum level. (**b**) Nominal molten aluminum level.

**Figure 6 sensors-23-05506-f006:**
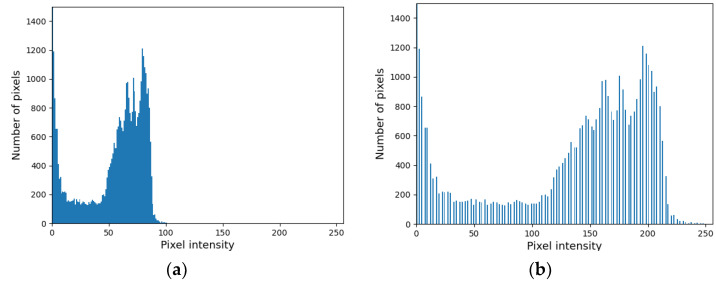
Histogram modification. (**a**) Original histogram translated to origin. (**b**) Histogram after stretching.

**Figure 7 sensors-23-05506-f007:**
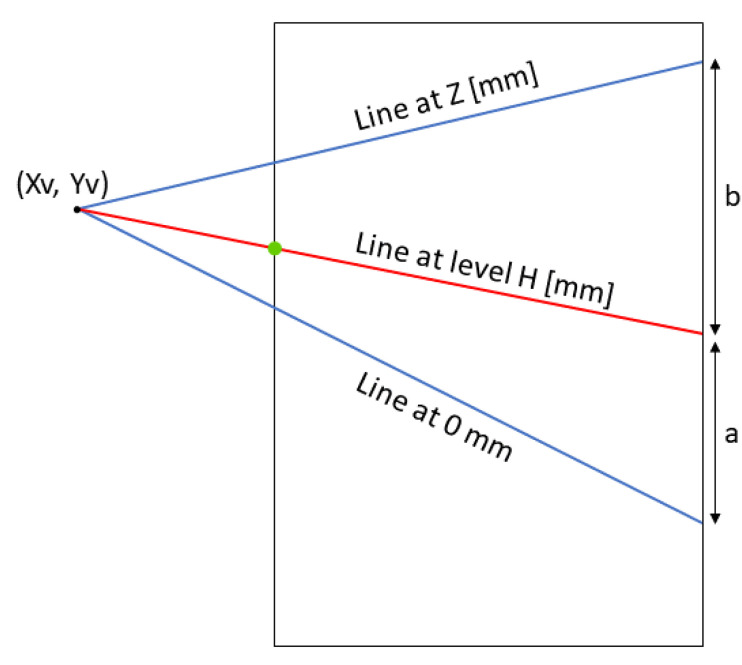
Illustration of the reference system; the box indicates the ROI of Figure 5; coordinates for the lines at 0 mm and Z (mm) are known from the furnace drawings; since lines are parallel in real world, they should all converge at the vanishing point (Xv, Yv).

**Figure 8 sensors-23-05506-f008:**
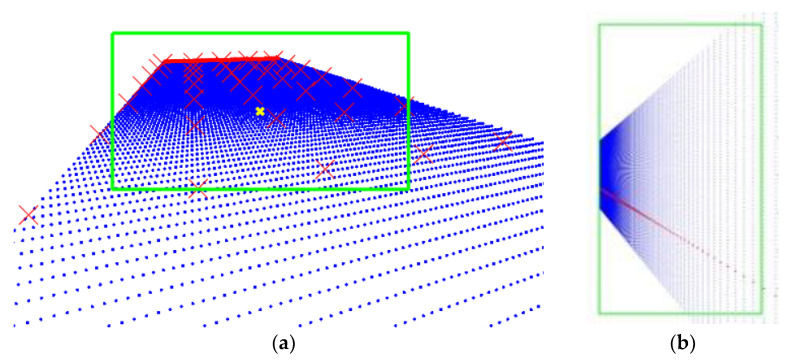
Camera field of view (FoV) simulated using raytracing algorithm. The green box represents the camera frame. (**a**) FoV of camera used for aluminothermic reaction identification. (**b**) FoV of camera used for melt level identification.

**Figure 9 sensors-23-05506-f009:**
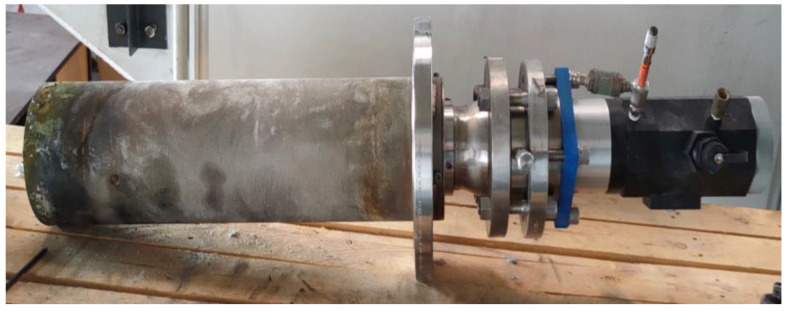
A picture of the camera system installed in the furnace.

**Figure 10 sensors-23-05506-f010:**
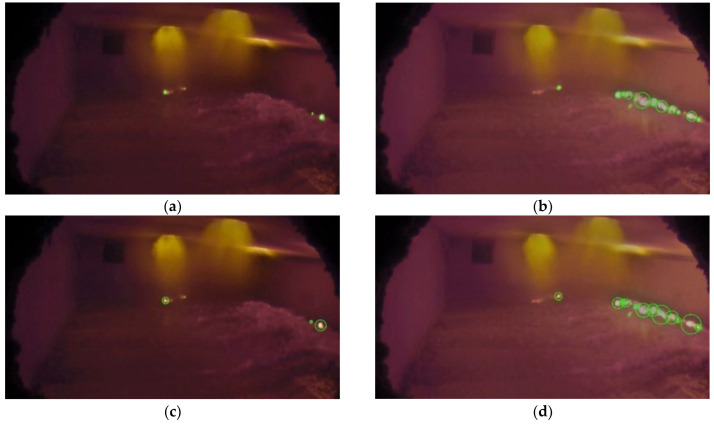
Aluminothermic reaction identification: (**a**) color scale algorithm identification for a low-rate reaction; (**b**) color scale algorithm identification for a medium-rate reaction; (**c**) grayscale algorithm identification for a low-rate reaction; and (**d**) grayscale algorithm identification for a medium-rate reaction.

**Figure 11 sensors-23-05506-f011:**
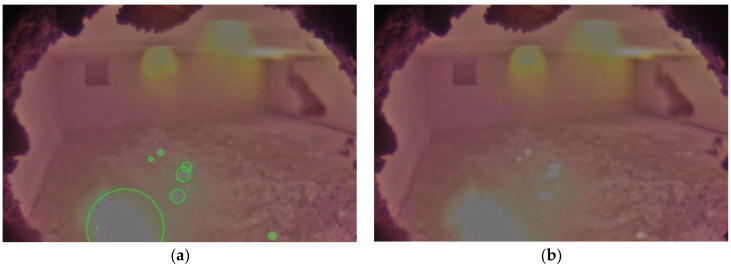
Aluminothermic reaction identification to a frame with high reaction rate: (**a**) identification using color scale algorithm; (**b**) identification using grayscale algorithm.

**Figure 12 sensors-23-05506-f012:**
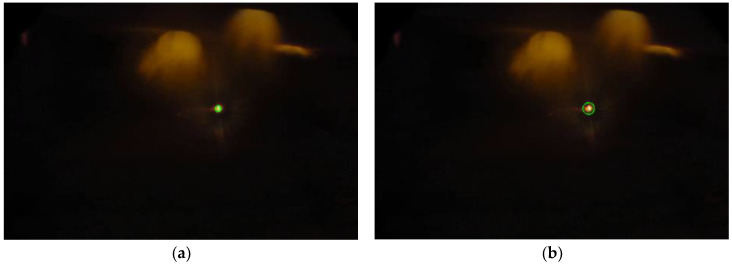
Identification of aluminothermic flames when the illumination is low. The reaction is at the initial stage as well. (**a**) Identification using color scale algorithm; (**b**) identification using grayscale algorithm.

**Figure 13 sensors-23-05506-f013:**
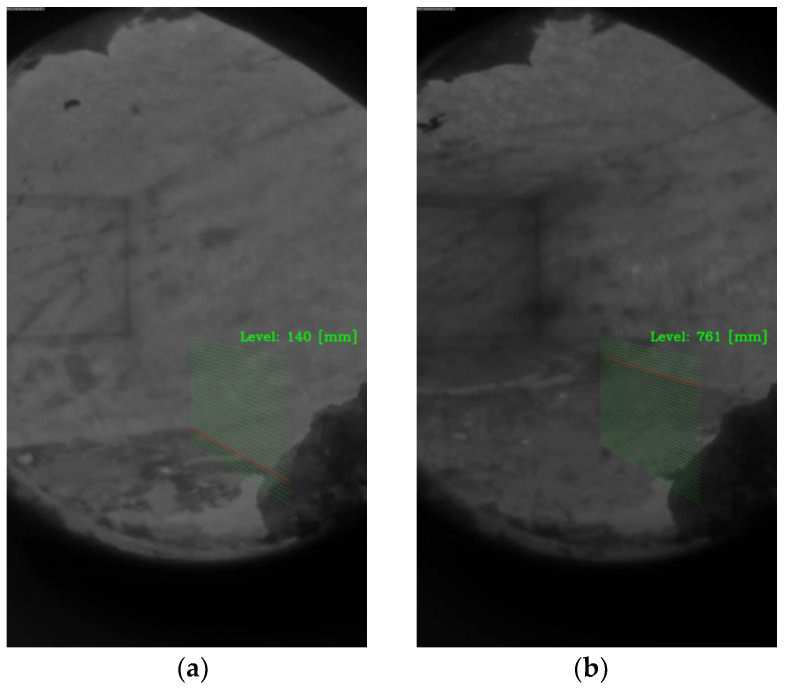
Level measurement for the detected line at (**a**) low molten aluminum level and (**b**) higher molten aluminum level.

**Figure 14 sensors-23-05506-f014:**
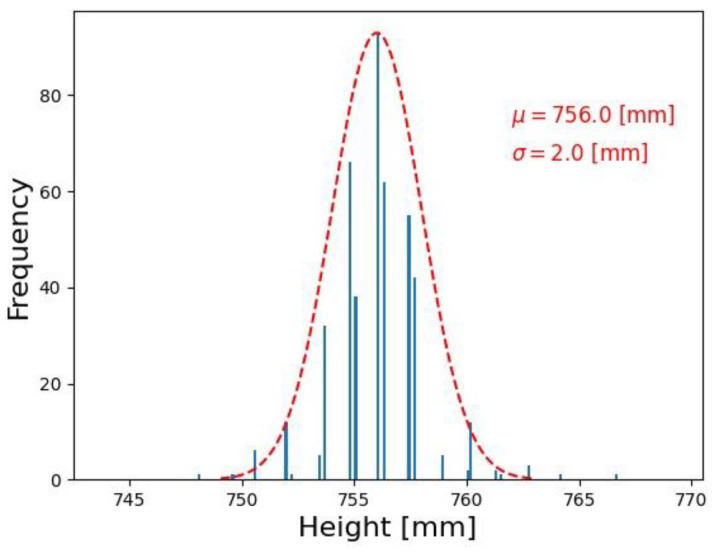
Distribution of the level measurement with mean (µ) and standard deviation (σ).

**Figure 15 sensors-23-05506-f015:**
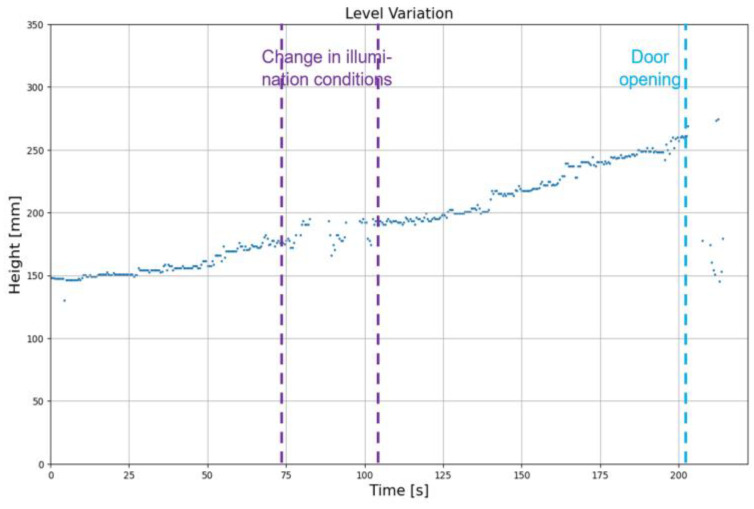
Measurements with rising melt levels and changing illumination conditions.

**Table 1 sensors-23-05506-t001:** Summary of the related literature with relevant contributions and limitations.

Reference	Relevant Contribution	Limitations
[8]	Provides a review of flame and smoke detection methods from the literature	[Not applicable]
[9]	Flame detection in videos using RGB color model	Detection performance is low when the flame is small, typically during the initial stage of fire
[10]	Fire detection in videos using color and motion information	The method developed for general data, such as video sequences taken in outdoor environments
[14]	Flame detection in video using a hidden Markov model	Videos were in outdoor environment with bright illumination
[16]	Explanation of color models	[Not applicable]
[18]	Matte levels were measured in a smelting furnace using magnetic sensors	Contact measurement method. A probe should be inserted into the molten bath
[20]	Molten steel level is measured by detecting the steel-flux interface using computer vision	The system requires a sounding bar and a mechanical system to periodically deliver the bar, making it a contact measurement method
[21]	Level measuring system using electromagnetic induction sensor and difference of magnetic field obtained	The sensor should be placed closer to the molten metal bath for an accurate measurement. The sensor electronics cannot withstand high-temperature furnace environment
[22]	Measurement of molten metal level in a furnace using an electromagnetic sensing system consisting of eddy current drive and pickup coils	Drive and coils should be embedded in the furnace walls. Accuracy of the measured level depends on the strength of the electromagnetic field
[23]	Molten metal level measured using a capacitance sensor system	The capacitance sensor should be placed closer to the molten metal bath for an accurate measurement
[25,26,27,28]	Automatic level measurement from images of liquid level with measuring scales or gauges	A level meter or gauge should be present to detect them using camera

**Table 2 sensors-23-05506-t002:** Performance metrics of the algorithms.

Algorithm	Average Recognition Rate	Average Cycle Time [s]
Color scale	1.99	0.69
Grayscale	6.51	0.18

## Data Availability

The data presented in this study are available on request. The data are not publicly available due to privacy restrictions.

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
