# Peer review of "Identification of Aluminothermic Reaction and Molten Aluminum Level through Vision System"

_sensors, 2023, doi:10.3390/s23125506_

Round 1
Reviewer 1 Report
This manuscript does an interesting demonstration on research progress of Identification of Aluminothermic Reaction and Molten Aluminium Level Through Vision System. Introduction is good however some similar approaches in literature need to be addressed. Very interesting presentation and very good information collection. During the reading of the manuscript, the following questions and comments came to my mind and I would like to ask the authors to comment on them:
1- Please check the file attached for details comments
2- In conclusion will be good if more results (actual finding supported by numbers) be presented than some general finding.
As the article provides a good body of work, I find that it has the sufficient quality to be published after some modifications.

Minor editing of English language required
Reviewer 2 Report
The article presents a detailed study of Identification of Aluminothermic Reaction and Molten Aluminium Level Through Vision System. The findings of this paper are significant and makes a valuable contribution to the field of study.
The following are the questions that can be shared with the authors and the manuscript should be revised accordingly.
1. A summary of the detailed literature study should be provided in the form of a table. The table should comprise the relevant contribution of the study and its limitation.
2. Initially full abbreviation should be used in the text followed by the acronyms.
3. The details about the data set is not provided in the manuscript. The number of the images taken and any preprocessing done in this regard should be explained in the revised manuscript.
4. The term/parameter confidence level has not been explained in the manuscript and only mentioned in the conclusion section.
5. Machine learning and deep learning techniques are not referred in the manuscript for the case study as it might provide even more accurate results.
6. How can aluminothermic reactions be compared to other types of chemical reactions in terms of efficiency and energy output?
7. What safety precautions should be taken when performing aluminothermic reactions?
8. What is the optimal combination of color recognition and pattern recognition algorithms for accurately identifying aluminothermic reactions using a vision system?
9. How does the detection accuracy of a vision system for aluminothermic reactions vary based on different lighting conditions, such as low light or bright light environments?
10. What is the feasibility of using a vision system for real-time monitoring and control of aluminothermic reactions in industrial applications, and what are the potential benefits and limitations of such an approach?
11. How can a vision system be integrated with other monitoring and control systems to optimize aluminothermic reaction processes and improve overall efficiency and safety?
12. How does the accuracy of aluminothermic reaction identification using a vision system compare to other non-invasive monitoring techniques, such as temperature sensing or pressure sensing?
13. Can a vision system be used to detect and identify other types of chemical reactions beyond aluminothermic reactions, and what are the potential applications and limitations of such an approach?
Reviewer 3 Report
Dear Authors,
nice and interesting paper,
and an interesting study.
It is always good to see some useful implementation of CV in extreme applications. (Excuse me, but I come from robotics, and 1000 Celsius is extreme for me).
I really like your idea, which is presented in a relatively easy-to-understand way, using an easy-to-follow language. Mostly. I have only a few language-related details:
line 17 - "algorithms (...) developed by processing (...)" - I am sure you do not mean "by" (like in "a song performed by Andrea Bocelli") because the "processing" would be the subject who does the job = the development of the "algorithms". Please ask a native how to fix this.
line 50 - "starting from the images" - I do not understand the idea of "starting". Why does the benefits start from the images? Please ask a native how to fix this.
line 50 - "starting from the images measured" - images can not be measured. They are data objects. They have some parameters, like filesize, resolution, colour-related parameters, and these can be quantified, and measured. After reading section 2.2.3 I think I know what you mean, but the reader should know it here and now, not after finishing reading the paper.
52 - "starting" - see 50
71 - "Grey scale" - who is Grey? What is his name? - I assume that you mean "grayscale", like in line 124 (?)
185 - grey scale - I assume that you mean "grayscale", with "a" and as one word, like in line 124 (?)
292,296,299,300,313,322,325,329,... - "Grey Scale" - I think that you try all of the combinations. Don't. Stick to the one and official "grayscale". And it is not a name, it can and should be written lowercase.
291 - "a" grayscale - well, you don't have a choice here. Well, you do, but you don't. You mean the grayscale = the opposite of colour = the lack of colour, so there is no "a" nor the "the" would be good here, while we can not point to "the" grayscale (as it would be a specific one (among other possibilities). Simply write "grayscale".
292,296,298,... - "Colour Scales" - although I like your idea of gathering numerous formats in one unofficial term, unfortunately, I can not agree that this is a name. It is not. It is a group of formats, not a name. So... lowercase.
330,367 - "0.37 s" - you should not allow such elements to be separated by a line break - instead of "0.37 s" write "0.37~s" as this will make a hard space that prevents splitting between lines.
370 - "Computer Vision" is a name of a big research area. Therefore it starts with capital letters.
I hope that you will be able to make good use of my insights and comments and accelerate your career and writing even more.
Best regards,
reviewer :)
Most of my above-mentioned comments relate to language.
However, I believe that the authors will manage to address all my concenrs without a problem.
Round 2
Reviewer 2 Report
It appears that the authors have made all the changes I have requested, and therefore their work can be accepted and published as it is currently written.